# Deliberation Networks: Sequence Generation Beyond One-Pass Decoding *

[1]**Yingce Xia**, [2]**Fei Tian**, [3]**Lijun Wu**, [1]**Jianxin Lin**, [2]**Tao Qin**, [1]**Nenghai Yu**, [2]**Tie-Yan Liu**
[1]University of Science and Technology of China, Hefei, China
[2]Microsoft Research, Beijing, China    [3]Sun Yat-sen University, Guangzhou, China
[1]yingce.xia@gmail.com, linjx@mail.ustc.edu.cn, ynh@ustc.edu.cn
[2]{fetia,taoqin,tie-yan.liu}@microsoft.com, [3]wulijun3@mail2.sysu.edu.cn

## Abstract

The encoder-decoder framework has achieved promising progress for many sequence generation tasks, including machine translation, text summarization, dialog system, image captioning, *etc*. Such a framework adopts an one-pass forward process while decoding and generating a sequence, but lacks the deliberation process: A generated sequence is directly used as final output without further polishing. However, deliberation is a common behavior in human's daily life like reading news and writing papers/articles/books. In this work, we introduce the deliberation process into the encoder-decoder framework and propose deliberation networks for sequence generation. A deliberation network has two levels of decoders, where the first-pass decoder generates a raw sequence and the second-pass decoder polishes and refines the raw sentence with deliberation. Since the second-pass deliberation decoder has global information about what the sequence to be generated might be, it has the potential to generate a better sequence by looking into future words in the raw sentence. Experiments on neural machine translation and text summarization demonstrate the effectiveness of the proposed deliberation networks. On the WMT 2014 English-to-French translation task, our model establishes a new state-of-the-art BLEU score of 41.5.

## 1 Introduction

The neural network based encoder-decoder framework has been widely adopted for sequence generation tasks, including neural machine translation [1], text summarization [19], image captioning [27], *etc*. In such a framework, the encoder encodes the source input $x$ with length $m$ into a sequence of vectors $\{h_1, h_2, \cdots, h_m\}$. The decoder, which is typically an RNN, generates an output sequence word by word[2] based on the source-side vector representations and previously generated words. The attention mechanism [1, 35], which dynamically attends to different parts of $x$ while generating each target-side word, is integrated into the encoder-decoder framework to improve the quality of generating long sequences [1].

Although the framework has achieved great success, one concern is that while generating one word, one can only leverage the generated words but not the future words un-generated so far. That is, when the decoder generates the $t$-th word $y_t$, only $y_{<t}$ can be used, while the possible words $y_{>t}$ are not explicitly considered. In contrast, in real-word human cognitive processes, global information, including both the past and the future parts, is leveraged in an iterative *polishing* process. Here are two examples: (1) Consider the situation that we are reading a sentence and meet an unknown word

---

[*]This work was done when Yingce Xia, Lijun Wu and Jianxin Lin were interns at Microsoft Research.
[2]Throughout this work, a *word* refers to the basic unit in a sequence.

in the middle of the sentence. We do not stop here. Instead, we move forward until the end of the sentence. Then we go back to the unknown word and try to understand it using its context, including the words both preceding and after it. (2) To write a good document (or paragraph, article), we usually first create a complete draft and then polish it based on global understanding of the whole draft. When polishing a specific part, we take the whole picture of the draft into consideration to evaluate how well the local element fits into the global environment rather than only looking back to the preceding parts.

We call such a polishing process as *deliberation*. Motivated by such human cognitive behaviors, we propose the *deliberation networks*, which leverage the global information with both looking back and forward in sequence decoding through a deliberation process. Concretely speaking, to integrate such a process into the sequence generation framework, we carefully design our architecture, which consists of two decoders, a first-pass decoder $\mathcal{D}_1$ and a second-pass/deliberation decoder $\mathcal{D}_2$, as well as an encoder $\mathcal{E}$. Given a source input $x$, the $\mathcal{E}$ and $\mathcal{D}_1$ jointly works like the standard encoder-decoder model to generate a coarse sequence $\hat{y}$ as a draft and the corresponding representations $\hat{s} = \{\hat{s}_1, \hat{s}_2, \cdots, \hat{s}_{T_{\hat{y}}}\}$ used to generate $\hat{y}$, where $T_{\hat{y}}$ is the length of $\hat{y}$. Afterwards, the deliberation decoder $\mathcal{D}_2$ takes $x, \hat{y}$ and $\hat{s}$ as inputs and outputs the refined sequence $y$. When $\mathcal{D}_2$ generates the $t$-th word $y_t$, an additional attention model is used to assign an adaptive weight $\beta_j$ to each $\hat{y}_j$ and $\hat{s}_j$ for any $j \in [T_{\hat{y}}]$, and $\sum \beta_j[\hat{y}_j; \hat{s}_j]$ is fed into $\mathcal{D}_2$.[3] In this way, the global information of the target sequence can be utilized to refine the generation process. We propose a Monte Carlo based algorithm to overcome the difficulty brought by the discrete property of $\hat{y}$ in optimizing the deliberation network.

To verify the effectiveness of our model, we work on two representative sequence generation tasks.

(1) *Neural machine translation* refers to using neural networks to translate sentences from a source language to a target language [1, 33, 32, 34]. A standard NMT model consists of an encoder (used to encode source sentences) and a decoder (used to generate target sentences), and thus can be improved by our proposed deliberation network. Experimental results show that based on a widely used single-layer GRU model [1], on the WMT'14 [29] English→French dataset, we can improve the BLEU score [17], by 1.7 points compared to the model without deliberation. We also apply our model on Chinese→English translations and improve BLEU by an averaged 1.26 points on four different test sets. Furthermore, on the WMT'14 English→French translation task, by applying deliberation to a deep LSTM model, we achieve a BLEU score 41.50, setting a new record for this task.

(2) *Text summarization* is a task that summarizes a long article into a short abstract. The encoder-decoder framework can also be used for such a task and thus could be refined by deliberation networks. Experimental results on Gigaword dataset [6] show that deliberation network can improve ROUGE-1, ROUGE-2, and ROUGE-L by 3.45, 1.70 and 3.02 points.

## 1.1  Related Work

Although there exist many works to improve the attention based encoder-decoder framework for sequence generation, such as changing the training loss [28, 18, 22] or the decoding objective [14, 7], not much attention has been paid to the structure of the encoder-decoder framework. Our work changes the structure of the framework by introducing the second-pass decoder into it.

The idea of deliberation/refinement is not well explored for sequence generation tasks, especially for the encoder-decoder based approaches [3, 23, 1] in neural machine translation. One related work is post-editing [16, 2]: a source sentence $e$ is first translated to $f'$, and then $f'$ is refined by another model. Different from our deliberation network, the two processes (i.e., generating and refining) in post-editing are separated. As a comparison, what we build is a consistent model in which all the components are coupled together and jointly optimized in an end-to-end way. As a result, deliberation networks lead to better accuracies. Another related work is the review network [36]. The idea is to review all the information encoded by the encoder to obtain thought vectors that are more compact and abstractive. The thought vectors are then used in decoding. Different from our work, the review steps are added on the encoder side, while the decoder side is unchanged and still adopts one-pass decoding.

The rest of our paper is organized as follows. Our proposed deliberation network is introduced in Section 2, including the model structure and the optimization process. Applications to neural machine translation and text summarization are introduced in Section 3 and Section 4 respectively. Section 5 concludes the paper and discusses possible future directions.

## 2 The Framework

In this section, we first introduce the overall architecture of deliberation networks, then the details of individual components, and finally propose an end-to-end Monte Carlo based algorithm to train the deliberation networks.

### 2.1 Structure of Deliberation Networks

As shown in Figure 1, a deliberation network consists of an encoder $\mathcal{E}$, a first-pass decoder $\mathcal{D}_1$ and a second-pass decoder $\mathcal{D}_2$. Deliberation happens at the second-pass decoder, which is also called deliberation decoder alternatively. Briefly speaking, $\mathcal{E}$ is used to encode the source sequence into a sequence of vector representations. $\mathcal{D}_1$ reads the encoder representations and generates a first-pass target sequence as a draft, which is further provided as input to the deliberation decoder $\mathcal{D}_2$ for the second-pass decoding. In the rest of this section, for simplicity of description, we use RNN as the basic building block for both the encoder and decoders[4]. All the $W$'s and $v$'s in this section with different superscripts or subscripts are the parameters to be learned. Besides, all the bias terms are omitted to increase readability.

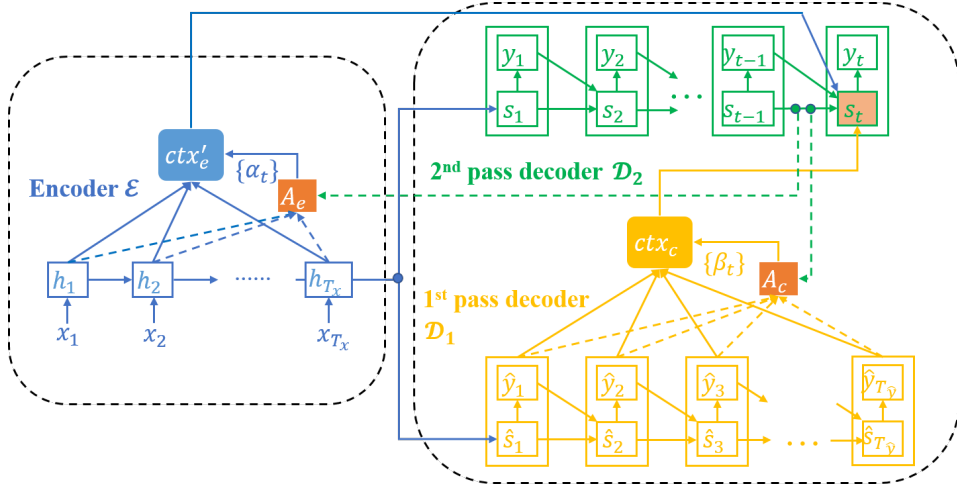

Figure 1: Framework of deliberation networks: Blue, yellow and green parts indicate encoder $\mathcal{E}$, first-pass decoder $\mathcal{D}_1$ and the second-pass decoder $\mathcal{D}_2$ respectively. The $\mathcal{E}$-to-$\mathcal{D}_1$ attention model is omitted for readability.

### 2.2 Encoder and First-pass Decoder

When an input sequence $x$ is fed into the encoder $\mathcal{E}$, it is encoded into $T_x$ hidden states $H = \{h_1, h_2, \cdots, h_{T_x}\}$ where $T_x$ is the length of $x$. Specifically, $h_i = \text{RNN}(x_i, h_{i-1})$, where $x_i$ acts as the representation (e.g., word embedding vector) for the $i$-th word in $x$ and $h_0$ is a zero vector.

The first-pass decoder $\mathcal{D}_1$ will generate a series of hidden states $\hat{s}_j \ \forall j \in [T_{\hat{y}}]$, and a first-pass sequence $\hat{y}_j \ \forall j \in [T_{\hat{y}}]$, where $T_{\hat{y}}$ is the length of the generated sequence. Next we show how they are generated in detail.

Similar to the conventional encoder-decoder model, an attention model is included in $\mathcal{D}_1$. At step $j$, the attention model in $\mathcal{D}_1$ first generates a context $ctx_e$ defined as follows:

$$ctx_e = \sum_{i=1}^{T_x} \alpha_i h_i;\ \alpha_i \propto \exp(v_\alpha^T \tanh(W_{att,h}^c h_i + W_{att,\hat{s}}^c \hat{s}_{j-1}))\, \forall i \in [T_x];\ \sum_{i=1}^{T_x} \alpha_i = 1. \quad (1)$$

Based on $ctx_e$, $\hat{s}_j$ is calculated as $\hat{s}_j = \text{RNN}([\hat{y}_{j-1}; ctx_e], \hat{s}_{j-1})$. After obtaining $\hat{s}_j$, another affine transformation is applied on the concatenated vector $[\hat{s}_j; ctx_e; \hat{y}_{j-1}]$. Finally, the results of the transformation are fed into a softmax layer, and the $\hat{y}_j$ is sampled out from the obtained multinomial distribution.

## 2.3 Second-Pass Decoder

Once the first-pass target sequence $\hat{y}$ is generated by the first-pass decoder $\mathcal{D}_1$, it is fed into the second-pass decoder $\mathcal{D}_2$ for further refinement. Based on the sequence $\hat{y}$ and the hidden states $\hat{s}_j$ $\forall j \in [T_{\hat{y}}]$ provided by $\mathcal{D}_1$, $\mathcal{D}_2$ eventually outputs the second-pass sequence $y$ via the deliberation process.

Specifically, at step $t$, $\mathcal{D}_2$ takes the previous hidden state $s_{t-1}$ generated by itself, previously decoded word $y_{t-1}$, the source contextual information $ctx'_e$ and the first-pass contextual information $ctx_c$ as inputs. Two detailed points are: (1) The computation of $ctx'_e$ is similar to that of $ctx_e$ shown in Eqn. (1) with two differences: First, $\hat{s}_{j-1}$ is replaced by $s_{t-1}$; second, the model parameters are different. (2) To obtain $ctx_c$, $\mathcal{D}_2$ has an attention model (i.e., the $A_c$ in Figure 1) that can map the words $\hat{y}_j$'s and the hidden states $\hat{s}_j$'s into a context vector. Mathematically speaking, in the refinement process at $t$-th time step, the first-pass contextual information vector $ctx_c$ is computed as:

$$ctx_c = \sum_{j=1}^{T_{\hat{y}}} \beta_j [\hat{s}_j; \hat{y}_j];\ \beta_j \propto \exp(v_\beta^T \tanh(W_{att,\hat{sy}}^d [\hat{s}_j; \hat{y}_j] + W_{att,s}^d s_{t-1}))\ \forall j \in [T_{\hat{y}}];\ \sum_{j=1}^{T_{\hat{y}}} \beta_j = 1.$$

As can be seen from the above computation, the deliberation process at time step $t$ in the second-pass decoding uses the whole sequence generated by the first-pass decoder, including both the words preceding and after $t$-th step in the first-pass sequence. That is, the first-pass contextual vector $ctx_c$ aggregates the global information extracted from the first-pass sequence $\hat{y}$.

After receiving $ctx_c$, we calculate $s_t$ as $s_t = \text{RNN}([y_{t-1}; ctx'_e; ctx_c], s_{t-1})$. Similar to sampling $\hat{y}_t$ in $\mathcal{D}_1$, $[s_t; ctx'_e; ctx_c; y_{t-1}]$ will be further transformed to generate $y_t$.

## 2.4 Algorithm

Let $D_{XY} = \{(x^{(i)}, y^{(i)})\}_{i=1}^n$ denote the training corpus with $n$ paired sequences[5]. Denote the parameters of $\mathcal{E}$, $\mathcal{D}_1$ and $\mathcal{D}_2$ as $\theta_e$, $\theta_1$ and $\theta_2$ respectively. The training of sequence-to-sequence learning is usually to maximize the data log likelihood $(1/n) \sum_{i=1}^n \log P(y_i|x_i)$. Under our setting, this rule can be specialized to maximize $(1/n) \sum_{(x,y) \in D_{XY}} \mathcal{J}(x, y; \theta_e, \theta_1, \theta_2)$, where

$$\mathcal{J}(x, y; \theta_e, \theta_1, \theta_2) = \log \sum_{y' \in \mathcal{Y}} P(y|y', E(x; \theta_e); \theta_2) P(y'|E(x; \theta_e); \theta_1). \quad (2)$$

In Eqn. (2), $\mathcal{Y}$ is the collection of all possible target sequences and $E(x; \theta_e)$ indicates a function that maps $x$ to its corresponding hidden states given by the encoder. One can verify that the first-order derivative of $\mathcal{J}(x, y; \theta_e, \theta_1, \theta_2)$ w.r.t $\theta_1$ is:

$$\nabla_{\theta_1} \mathcal{J}(x, y; \theta_e, \theta_1, \theta_2) = \frac{\sum_{y' \in \mathcal{Y}} P(y|y', E(x; \theta_e); \theta_2) \nabla_{\theta_1} P(y'|E(x; \theta_e); \theta_1)}{\sum_{y' \in \mathcal{Y}} P(y|y', E(x; \theta_e); \theta_2) P(y'|E(x; \theta_e); \theta_1)},$$

which is extremely hard to compute due to the large space of $\mathcal{Y}$. Similarly, the gradients w.r.t. $\theta_e$ and $\theta_2$ are also computationally intractable. To overcome such difficulties, we propose a Monte Carlo based method to optimize the lower bound of $\mathcal{J}(x, y; \theta_e, \theta_1, \theta_2)$. Note by the concavity of $\mathcal{J}$ w.r.t $y'$, one can verify that $\mathcal{J}(x, y; \theta_e, \theta_1, \theta_2) \geq \tilde{\mathcal{J}}(x, y; \theta_e, \theta_1, \theta_2)$, with the right-hand side acting as a lower bound and defined as

$$\tilde{\mathcal{J}}(x, y; \theta_e, \theta_1, \theta_2) = \sum_{y' \in \mathcal{Y}} P(y'|E(x; \theta_e); \theta_1) \log P(y|y', E(x; \theta_e); \theta_2). \quad (3)$$

Denote $\tilde{\mathcal{J}}(x,y;\theta_e,\theta_1,\theta_2)$ as $\tilde{J}$. The gradients of $\tilde{J}$ w.r.t its parameters are:

$$\nabla_{\theta_1}\tilde{\mathcal{J}} = \sum_{y'\in\mathcal{Y}} P(y'|E(x;\theta_e);\theta_1)\underbrace{\log P(y|y',E(x;\theta_e);\theta_2)\nabla_{\theta_1}\log P(y'|E(x;\theta_e);\theta_1)}_{G_1};$$

$$\nabla_{\theta_2}\tilde{\mathcal{J}} = \sum_{y'\in\mathcal{Y}} P(y'|E(x;\theta_e);\theta_1)\underbrace{\nabla_{\theta_2}\log P(y|y',E(x;\theta_e);\theta_2)}_{G_2}; \quad (4)$$

$$\nabla_{\theta_e}\tilde{\mathcal{J}} = \sum_{y'\in\mathcal{Y}} P(y'|E(x;\theta_e);\theta_1)G_e(x,y,y';\theta_e,\theta_1,\theta_2), \text{ where } G_e \text{ is defined as follows:}$$

$$G_e = \nabla_{\theta_e}\log P(y|y',E(x;\theta_e);\theta_2) + \log P(y|y',E(x;\theta_e);\theta_2)\nabla_{\theta_e}\log P(y'|E(x;\theta_e);\theta_1).$$

Let $\Theta = [\theta_1;\theta_2;\theta_e]$ and $G(x,y,y';\Theta) = [G_1;G_2;G_e]$, where $G_1$, $G_2$ and $G_e$ are defined in Eqn. (4). (For ease of reference, we assume that all the $\theta$.'s and $G$.'s are flattened.) Obviously, if $y'$ is sampled from distribution $P(y'|E(x;\theta_e);\theta_1)$, $G(x,y,y';\Theta)$ is an unbiased estimator of the gradient of $\tilde{J}$ w.r.t. all model parameters $\Theta$. Based on that we propose our algorithm in Algorithm 1.

---

**Algorithm 1:** Algorithm to train the deliberation network

---

**Input**: Training data corpus $D_{XY}$; minibatch size $m$; optimizer $Opt(\cdots)$ with gradients as input ;
**while** *models not converged* **do**

> Randomly sample a mini-batch of $m$ sequence pairs $\{x^{(i)},y^{(i)}\}\ \forall i\in[m]$ from $D_{XY}$;
> For any $x^{(i)}$ where $i\in[m]$, sample $y'^{(i)}$ according to distribution $P(\cdot|E(x^{(i)};\theta_e);\theta_1)$;
> Perform parameter update: $\Theta \leftarrow \Theta + Opt(\frac{1}{m}\sum_{i=1}^{m} G(x^{(i)},y^{(i)},y'^{(i)};\Theta))$.

---

**Discussions** (1) The choice of *Opt*(...) is quite flexible. One can choose different optimizers such as Adadelta [37], Adam [13], or SGD for different tasks, depending on common practice in the specific task. (2) The $\mathcal{Y}$ space is usually extremely large in sequence generation tasks. To obtain better sampled $y'$, we can use beam search instead of randomly sampling.

## 3 Application to Neural Machine Translation

We evaluate the deliberation networks with two different network structures: (1) the shallow model, which is based on a widely-used single-layer GRU model named *RNNSearch* [1, 12]; (2) the deep model, which is based on a deep LSTM model similar to *GNMT* [31]. Both of the two kinds of models are implemented in Theano [24].

### 3.1 Shallow Models

#### 3.1.1 Settings

*Datasets* We work on two translation tasks, English-to-French translation (denoted as En→Fr) and Chinese-to-English translation (denoted as Zh→En). For En→Fr, we employ the standard filtered WMT'14 dataset[6], which is widely used in NMT literature [1, 12]. There are 12M bilingual sentence pairs in the dataset. We concatenate *newstest2012* and *newstest2013* together as the validation set and use *newstest2014* as the test set. For Zh→En, we choose 1.25M bilingual sentence pairs from LDC dataset as training corpus, use NIST2003 as the validation set, and NIST2004, NIST2005, NIST2006, NIST2008 as the test sets. Following the common practice [1, 12], we remove the sentences with more than 50 words for both translation tasks. Furthermore, we limit the both the source words and target words as $30k$ most-frequent ones. The out-of-vocabulary words are replaced by a special token "UNK".

*Model* We choose the most widely adopted NMT model RNNSearch [1, 12, 25] as the basic structure to construct the deliberation network. To be specific, all of $\mathcal{E}$, $\mathcal{D}_1$ and $\mathcal{D}_2$ are GRU networks [1] with one hidden layer of 1000 neurons. The word embedding dimension is set as 620. For Zh→En, we apply 0.5 dropout rate to the layer before softmax and no dropout is used in En→Fr translation.

*Optimization* All the models are trained on a single NVIDIA K40 GPU. We first pre-train two standard encoder-decoder based NMT models (i.e., RNNSearch) until convergence, which take about two weeks for En→Fr and one week for Zh→En using Adadelta [37]. For any deliberation network, (1) the encoder is initialized by the encoder of the pre-trained RNNSearch model; (2) both the first-pass and second-pass decoders are initialized by the decoder of the pre-trained RNNSearch model; (3) the attention model used to compute the first-pass context vector is randomly initialized from a uniform distribution on $[-0.1, 0.1]$. Then we train the deliberation networks by Algorithm 1 until convergence, which takes roughly 5 days for both tasks. The minibatch size is fixed as 80 throughout the optimization. Plain SGD is used as the optimizer in this process, with initial learning rate $0.2$ and halving according to validation accuracy. To sample the intermediate translation output by the first decoder, we use beam search with beam size 2, considering the tradeoff between accuracy and efficiency.

*Evaluation* We use BLEU [17] as the evaluation metric for translation qualities. BLEU is the geometric mean of $n$-gram precisions where $n \in \{1, 2, 3, 4\}$, weighted by sentence lengths. Following the common practice in NMT, we use *multi-bleu.pl*[7] to calculate case-sensitive BLEU scores for En→Fr, while evaluating the translation qualities of Zh→En by case-insensitive BLEU scores. The larger the BLEU score is, the better the translation quality is. For the baselines and deliberation networks, we use beam search with beam size 12 to generate sentences.

*Baselines* We compare our proposed algorithms with the following baselines: (i) The standard NMT algorithm RNNSearch [1, 12], denoted as $\mathcal{M}_{\text{base}}$; (ii) The standard NMT model with two stacked decoding layers, denoted as $\mathcal{M}_{\text{dec}\times 2}$; (3) The review network proposed in [36]. We try both 4 and 8 reviewers and find the 4-reviewer model is slightly better. The review network in our experiment is therefore denoted as $\mathcal{M}_{\text{reviewer}\times 4}$. We refer to our proposed algorithm as $\mathcal{M}_{\text{delib}}$.

### 3.1.2 Results

We compare our proposed algorithms with the following baselines: (i) The standard NMT algorithm, denoted as $\mathcal{M}_{\text{base}}$; (ii) The standard NMT model with two stacked decoding layers, denoted as $\mathcal{M}_{\text{dec}\times 2}$; (3) The review network proposed in [36]. We try both 4 and 8 reviewers and find the 4-reviewer model is slightly better. The review network in our experiment is therefore denoted as $\mathcal{M}_{\text{reviewer}\times 4}$. We refer to our proposed algorithm as $\mathcal{M}_{\text{delib}}$. Table 1 shows the results of En→Fr translation. We have several observations:

(1) Our proposed algorithm performs the best among all candidates, which validates the effectiveness of the deliberation process. (2) Our method $\mathcal{M}_{\text{delib}}$ outperforms the baseline algorithm $\mathcal{M}_{\text{base}}$. This shows that further polishing the raw output indeed leads to better sequences. (3) Applying an additional decoding layer, i.e., $\mathcal{M}_{\text{dec}\times 2}$, increases the translation quality, but it is still far behind that of $\mathcal{M}_{\text{delib}}$. Clearly, the second decoder layer of $\mathcal{M}_{\text{dec}\times 2}$ can still only leverage the previously generated words but not unseen and un-generated future words, while the second-pass decoder of $\mathcal{M}_{\text{delib}}$ can leverage the richer information contained in all the words from the first-pass decoder. Such a refinement process from the global view significantly improves the translation results. (4) $\mathcal{M}_{\text{delib}}$ outperforms $\mathcal{M}_{\text{reviewer}\times 4}$ by $0.91$ point, which shows that reviewing the possible future contextual information from the source side is not enough. The "future" information from the decoder side is also very important, since it is directly related with the final output.

Table 1: BLEU scores of En→Fr translation

| Algorithm | $\mathcal{M}_{\text{base}}$ | $\mathcal{M}_{\text{dec}\times 2}$ | $\mathcal{M}_{\text{reviewer}\times 4}$ | $\mathcal{M}_{\text{delib}}$ |
|---|---|---|---|---|
| BLEU | 29.97 | 30.40 | 30.76 | **31.67** |

The translation results of Zh→En are summarized in Table 2. We have similar observations as those for En→Fr translations: $\mathcal{M}_{\text{delib}}$ outperforms all the baseline methods, particularly with an average gain of $1.26$ points over $\mathcal{M}_{\text{base}}$.

Apart from the quantitative analysis, we list two examples in Table 3 to better understand how a deliberation network works. Each example contains five sentences, which are the source sentence in Chinese, the reference sentence in English as ground truth translation, the translation generated

Table 2: BLEU scores of Zh→En translation

| Algorithm | NIST04 | NIST05 | NIST06 | NIST08 |
|-----------|--------|--------|--------|--------|
| $\mathcal{M}_{\text{base}}$ | 34.96 | 34.57 | 32.74 | 26.21 |
| $\mathcal{M}_{\text{delib}}$ | 36.90 | 35.57 | 33.90 | 27.13 |

by $\mathcal{M}_{\text{base}}$ and the output translation by both the first-pass decoder and second-pass decoder (i.e., the final translation by deliberation network $\mathcal{M}_{\text{delib}}$).

Table 3: Case studies of Zh→En translations. Note the "......" in the second example represents a common sentence "*the two sides will discuss how to improve the implementation of the cease-fire agreement*".

> [Source] ***Aiji shuo, zhongdong heping*** *xieyi yuqi jiang you yige xinde jiagou* .
> [Reference] ***Egypt says*** *a new framework is expected to come into being for the* ***Middle East***
> *peace agreement* .
> [Base] ***egypt 's middle east*** *peace agreement is expected to have a new framework , he said* .
> [First-pass] ***egypt 's middle east*** *peace agreement is expected to have a new framework , egypt said* .
> [Second-pass] ***egypt says the middle*** *east peace agreement is expected to have a new framework* .

> [Source] *Nuowei dashiguan zhichu, "shuangfang jiang taolun ruhe gaijin luoshi tinghuo xieyi, zhe*
> *yeshi* ***san nian lai*** *shuangfang shouci* ***zai ruci gao de cengji shang jinxing mianduimian tanpan"***
> [Reference] *The Norwegian embassy pointed out that , " Both sides will discuss how to improve the*
> *implementation of the cease-fire agreement , which is the first time for both sides to* ***have***
> ***face-to-face negotiations at such a high level*** *. "*
> [Base] *" ...... , which is the first time for the two countries to* ***conduct face-to-face talks on the basis***
> ***of a high level of three years*** *, " it said* .
> [First-pass] *" ...... , which is the first time for the two countries to* ***conduct face-to-face talks on the***
> ***basis of a high level of three years*** *, " the norwegian embassy said in a statement* .
> [Second-pass] *" ...... , which is the first time* ***in three years*** *for the two countries to* ***conduct***
> ***face-to-face talks at such high level*** *, " the norwegian embassy said* .

In the first example, the translation from both base model and first-pass decoder contains the phrase *egypt's middle east peace agreement*, which is odd and inaccurate, given that an agreement cannot belong to a single country as Egypt. As a comparison, the second-pass decoder refines such phrase into a more natural and accurate one. i.e., *egypt says the middle east peace agreement*, by looking forward to the future translation phrase "*egypt said*" output by the first-pass decoder. On the other hand, the second-pass decoder outputs a sentence with correct tense, i.e., *egypt **says** ... **is** ....* However, the two sentences output by $\mathcal{M}_{\text{base}}$ and the first-pass decoder are inconsistent in tense, whose structures are "*... **is** ..., egypt **said** *". This problem is well addressed by the deliberation network, since the second-pass decoder can access the global information contained in the draft sequence generated by the first-pass decoder, and therefore output a more consistent sentence.

In the second example, as shown in bold fonts, the phrase "*conduct face-to-face talks on the basis of a high level of three years*" from both base model and first-pass decoder carries all necessary information of its corresponding source segments, but apparently it is out-of-order and seems to be a simple combination of words. The second-pass decoder refines such translation into a correct, and more fluent one, by forwarding the sub phrase *in three years* to the position right after *the first time*.

At last we compare the decoding time of deliberation network with that of the RNNSearch. Based on the Theano implementation, to translate 3003 English sentences to French, RNNSearch takes 964 seconds while the deliberation network takes 1924 seconds. Indeed, the deliberation network takes roughly 2 times decoding time of RNNSearch, but can bring 1.7 points improvements in BLEU.

## 3.2 Deep Models

We work on a deep LSTM model to further evaluate deliberation networks through the WMT'14 En→Fr translation task. Compared to the shallow model, there are several different aspects: (1) We use 34M sentence pairs from WMT'14 as training data, apply the BPE [21] techniques to split the training sentences into sub-word units and restrict the source and target sentence lengths within 64 subwords. The encoder and decoder share a common vocabulary containing $36k$ subwords. (2) All of $\mathcal{E}$, $\mathcal{D}_1$ and $\mathcal{D}_2$ are 4-layer LSTMs with residual connections [9, 10]. The word embedding dimension

Table 4: Comparison between deliberation network and different deep NMT systems (En→Fr).

| System | Configurations | BLEU |
|---|---|---|
| GNMT [31] | Stacked LSTM (8-layer encoder + 8 layer decoder) + RL finetune | 39.92 |
| FairSeq [4] | Convolution (15-layer) encoder and (15-layer) decoder | 40.51 |
| Transformer [26] | Self-Attention + 6-layer encoder + 6-layer decoder | 41.0 |
| *this work* | Stack LSTM (4-layer encoder and 4-layer decoder) | 39.51 |
| | Stack 4-layer NMT + Dual Learning | 40.53 |
| | Stack 4-layer NMT + Dual Learning + Deliberation Network | **41.50** |

and hidden node dimension are 512 and 1024 respectively. The dropout rate is set as $0.1$. (3) We train the standard encoder-decoder based deep model for about 25 days until convergence. Furthermore, we leverage our recently proposed *dual learning* techniques [8, 33] to improve the model, which takes another 7 days. We initialize the deliberation network in the same way in Section 3.1.1. Then, we train the deliberation network by Algorithm 1 for 10 days. When generating translations, we use beam search with beam size 8.

The experimental results of applying deliberation network to the deep LSTM model are shown in Table 4. On En→Fr translation task, the baseline of our implemented NMT system is 39.51. With dual learning, we achieve a 40.53 BLEU score. After applying deliberation techniques, the BLEU score can be further improved to 41.50, which as far as we know, is a new single-model state-of-the-art result for this task. This not only illustrates the effectiveness of deliberation network again, but also shows that even if a model is good enough, it can still benefit from the deliberation process.

## 4 Application to Text Summarization

We further verify the effectiveness of deliberation networks on text summarization, which is another real-world application that encoder-decoder framework succeeds to help [19].

### 4.1 Settings

Text summarization refers to using a short and abstractive sentence to summarize the major points of a sentence or paragraph, which is typically much longer. The training, validation and test sets for the task are extracted from Gigaword Corpus [6]: For each selected article, the first sentence is used as source-side input and the title used as target-side output. We process the data in the same way as that proposed in [20, 30], and obtain training/validation/test sets with roughly $189k/18k/10k$ sentence pairs respectively. There are roughly $42k$ unique words in the source input and $19k$ unique words in the target output and we remain all of them as the vocabulary in the encoder-decoder models.

The model structure is the same as that used in Section 3.1 except that both word embedding dimension and hidden node size are reduced to 128. We use Adadelta algorithm with gradient clip value $5.0$ to optimize deliberation network. The mini-batch size is fixed as 32.

The evaluation measures are chosen as ROUGE-1, ROUGE-2 and ROUGE-L, which are all widely adopted evaluation metric for text summarization [15]. ROUGE-$N$ ($N = 1, 2$ in our setting) is an N-gram recall between a candidate summary and a set of reference summaries. ROUGE-L is a similar statistic like ROUGE-$N$ but based on longest common subsequences. When generating the titles, we use beam search with beam size 10. For the thoroughness of comparison, similar to NMT, we add another two baselines apart from the basic encoder-decoder model: the stacked-decoder model with 2 layers ($\mathcal{M}_{\text{dec}\times 2}$), as well as the review net with 4 reviewers ($\mathcal{M}_{\text{reviewer}\times 4}$).

### 4.2 Results

The experimental results of text summarization are listed in Table 5. Again, the deliberation network achieves clear improvements over all the baselines. For example, in terms of ROUGE-2, it is 1.12 and 0.96 points better compared with stacked decoder model and review net respectively. Furthermore, one may note that a significant difference between NMT and text summarization is that: In NMT, the lengths of input and output sequence are very close; but in text summarization, the input is extremely

long while the output is very short. The better results brought by deliberation networks shows that even if the output sentence is short, it is helpful to include the deliberation process which refines the low-level draft in the first-pass decoder.

Table 5: ROUGE-{1, 2, L} scores of text summarization

| Algorithm | ROUGE-1 | ROUGE-2 | ROUGE-L |
|---|---|---|---|
| $\mathcal{M}_{\text{base}}$ | 27.45 | 10.51 | 26.07 |
| $\mathcal{M}_{\text{dec}\times 2}$ | 27.93 | 11.09 | 26.50 |
| $\mathcal{M}_{\text{reviewer}\times 4}$ | 28.26 | 11.25 | 27.28 |
| $\mathcal{M}_{\text{delib}}$ | 30.90 | 12.21 | 29.09 |

## 5   Conclusions and Future Work

In this work, we have proposed deliberation networks for sequence generation tasks, in which the first-pass decoder is used for generating a raw sequence, and the second-pass decoder is used to polish the raw sequence. Experiments show that our method achieves much better results than several baseline methods in both machine translation and text summarization, and achieves a new single model state-of-the-art result on WMT'14 English to French translation.

There are multiple promising directions to explore in the future. First, we will study how to apply the idea of deliberation to tasks beyond sequence generation, such as improving the image qualities generated by GAN [5]. Second, we will study how to refine/polish different levels of a neural network, like the hidden states in an RNN, or feature maps in a CNN. Third, we are curious about whether better sequences can be generated with more passes of decoders, i.e., refining a generated sequence again and again. Fourth, we will study how to speed up the inference of deliberation networks and reduce their inference time.

**Acknowledgments**

The authors would like to thank Yang Fan and Kaitao Song for implementing the deep neural machine translation basic model. This work is partially supported by the National Natural Science Foundation of China (Grant No. 61371192).

## Footnotes

[3]In this work, let $[v_1; v_2; \cdots; v_n]$ denote the long vector concatenated by the input vectors $v_1, \cdots, v_n$. With a little bit confusion, $[m]$ with a single integer input $m$ denotes the set $\{1, 2, \cdots, m\}$.

[4]The proposed deliberation networks are independent to the specific implementation of the recurrent units and can be applied to simple RNN or its variants such as LSTM [11] or GRU [3].

[5]Let $x^{(i)}$ and $y^{(i)}$ denote $i$'th source input and target output in the training data. Let $x_i$ and $y_i$ denote the $i$-th word in $x$ and $y$.

[6] http://www-lium.univ-lemans.fr/~schwenk/cslm_joint_paper/data/bitexts.tgz

[7]https://github.com/moses-smt/mosesdecoder/blob/master/scripts/generic/multi-bleu.perl

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
