[Reviews · NeurIPS 2017]

Reviewer 1



This paper introduces a two steps decoding model applied to NMT and summarization. The idea is well explained and the experimental results show significant improvements over realistic baseline. The authors chose to approximate the marginalization of the intermediate hypothesis with monte-carlo. This is indeed a solution, but it is not clear how many samples are used? Does this hyperparameter have an impact on the performance ? Do you use beam search instead of simple sampling? These technical details should be more precisely described. At the end, the work described in this paper is interesting and the experimental part is meaningful. However, I found the introduction very pretentious. Two paragraphs are dedicated to a "cognitive" justification as the authors have reinvented the wheel. Multi-steps decoding exists for many years and this is really a great idea to propose a solution for end to end neural model, but you don't need to discard previous work or motivate it with pop cognitive consideration (note that in "cognitive science" there is "science"). Page 2: The sentence in line 73 is useless. Line 78: this sentence is correct for end to end neural models, but there is a long history in speech recognition with multiple pass decoding: in the first step, a word lattice is generated with "weak" acoustic and language models, then more sophisticated and powerful models are used to refine iteratively this word lattice. So you cannot say that for sequence prediction it wasn't explored. It is not mandatory to discard previous work because you find a nice name. Page 8: Line 277: "In this work, inspired by human cognitive process," You could avoid this kind of pretentious phrase.

Reviewer 2



----- After response: thank you for the thorough response. Two of my major concerns: the weakness of the baseline and the lack of comparison with automatic post-editing have been resolved by the response. I've raised my evaluation with the expectation that these results will be added to the final camera ready version. With regards to the examples, the reason why I said "cherry-picked?" (with a question mark) was because there was no mention of how the examples were chosen. If they were chosen randomly or some other unbiased method that could be noted in the paper. It's OK to cherry-pick representative examples, of course, and it'd be more clear if this was mentioned as well. Also, if there was some quantitative analysis of which parts of the sentence were getting better this would be useful, as I noted before. Finally, the explanation that beam search has trouble recovering from search errors at the beginning of the sentence is definitely true. However, the fact that this isn't taken into account at training and test time is less clear. NMT is maximizing the full-sentence likelihood, which treats mistakes at any part of the sentence equally. I think it's important to be accurate here, as this is a major premise of this work. --------- This paper presents a method to train a network to revise results generated by a first-pass decoder in neural sequence-to-sequence models. The second-level model is trained using a sampling-based method where outputs are sampled from the first-pass model. I think idea in the paper is potentially interesting, but there are a number of concerns that I had about the validity and analysis of the results. It would also be better if the content could be put in better context of prior work. First, it wasn't entirely clear to me that the reasoning behind the proposed method is sound. There is a nice example in the intro where a poet used the final word of the sentence to determine the first word, but at least according to probabilities, despite the fact that standard neural sequence-to-sequence models are *decoding* from left to right, they are still calculating the joint probability over words in the entire sentence, and indirectly consider the words in the end of the sentence in the earlier decisions because the conditional probability of latter words will be lower if the model makes bad decisions for earlier words. Of course this will not work if greedy decoding is performed, but with beam search it is possible to recover. I think this should be discussed carefully. Second, regarding validity of the results: it seems that the baselines used in the paper are not very representative of the state of the art for the respective tasks. I don't expect that the every paper has to be improving SOTA numbers, but for MT if "RNNSearch" is the original Bahdanau et al. attentional model, it is likely missing most of the improvements that have been made to neural MT over the past several years, and thus it is hard to tell how these results will transfer to a stronger model. For example, the below paper achieves single-model BLEU scores on the NIST04-06 data sets that are 4-7 BLEU points higher than the proposed model: > Deep Neural Machine Translation with Linear Associative Unit. Mingxuan Wang, Zhengdong Lu, Jie Zhou, Qun Liu (ACL 2017) Third, the paper has a nominal amount of qualitative evaluation of (cherry-picked?) examples, but lacks any analysis of why the proposed method is doing better. It would be quite useful if analysis could be done to show whether the improvements are indeed due to the fact that the network has access to future target words in a quantitative fashion. For example, is the proposed method better at generating beginnings of sentences, where standard models are worse at generating words in this more information-impoverished setting. Finally, there is quite a bit of previous work that either is similar methodologically or attempts to solve similar problems. The closest is automatic post-editing or pre-translation such as the following papers: > Junczys-Dowmunt, Marcin, and Roman Grundkiewicz. "Log-linear combinations of monolingual and bilingual neural machine translation models for automatic post-editing." WMT2016. > Niehues, Jan, et al. "Pre-Translation for Neural Machine Translation." COLING2016 This is not trained jointly with the first-past decoder, but even without joint training is quite useful. Also, the following paper attempts to perform globally consistent decoding, and introduces bilingual or bidirectional models that can solve a similar problem: > Hoang, Cong Duy Vu, Gholamreza Haffari, and Trevor Cohn. "Decoding as Continuous Optimization in Neural Machine Translation." arXiv 2017.

Reviewer 3



This paper proposes a two stage decoding for Seq2Seq models: first, an output sequence is generated, and then, a second improved target sequence is produced using attention on the source and this initially created target sequence. The motivation behind is that this procedure would help to see the whole target sequence when making decision about word choice, etc. Although the authors show some improvements in the BLEU score by this method, I find the overall approach somehow awkward. Seq2Seq models have replaced traditional SMT, eg. phrase-based models (PBSMT) because it's a nice end-to-end framework with a global training objective, instead of many individually models which were first trained independently and then combined in a log-linear framework (PBSMT). You basically train a second NMT model which translates from a 1st target hypothesis to an improved one. This is similar in spirit to "statistical post-editing (SPE)". This is computationally expensive (+50% training time, decode 2 times slower). Do you use beam search for both decoder networks ? I would also say that your baseline is quite old: RNNSearch was the first model for NMT. In the meantime, better architectures are known and it would be good to show that the proposed two-stage approach improves on top of those systems (you mention yourself deeper encoder/decoder architectures). An alternative, easy to implement baseline, would be to perform a forward and backward decode (last to 1st word) and combine both.